# High-mobility group box (TOX) antibody a useful tool for the identification of B and T cell subpopulations

Lorena Maestre[1], Juan Fernando García-García[2], Scherezade Jiménez[1], Ana Isabel Reyes-García[1], Álvaro García-González[1], Santiago Montes-Moreno[3], Alberto J. Arribas[4], Patricia González-García[5], Eduardo Caleiras[5], Alison H. Banham[6], Miguel Ángel Piris[7]ᵒ, Giovanna Roncador[1]ᵒ*

1 Monoclonal Antibodies Core Unit, CNIO, Madrid, Spain, 2 Department of Pathology, MD Anderson Cancer Center Madrid, Madrid, Spain, 3 Hospital Universitario Marqués de Valdecilla, Pathology Department, Santander, Spain, 4 Università della Svizzera Italiana, Institute of Oncology Research, Bellinzona, Switzerland, 5 Histopathology Core Unit, CNIO, Madrid, Spain, 6 Nuffield Division of Clinical Laboratory Sciences, Radcliffe Department of Medicine, University of Oxford, Oxford, United Kingdom, 7 Department of Pathology, Fundación Jiménez Díaz, CIBERONC, Madrid, Spain

ᵒ These authors contributed equally to this work.
* groncador@cnio.es

**Data Availability Statement:** All relevant data are within the manuscript and its Supporting Information files.

## Abstract

Thymocyte selection-associated high-mobility group box (TOX) is a DNA-binding factor that is able to regulate transcription by modifying local chromatin structure and modulating the formation of multi-protein complexes. TOX has multiple roles in the development of the adaptive immune system including development of CD4 T cells, NK cells and lymph node organogenesis. However very few antibodies recognizing this molecule have been reported and no extensive study of the expression of TOX in reactive and neoplastic lymphoid tissue has been performed to date. In the present study, we have investigated TOX expression in normal and neoplastic lymphoid tissues using a novel rat monoclonal antibody that recognizes its target molecule in paraffin-embedded tissue sections. A large series of normal tissues and B- and T-cell lymphomas was studied, using whole sections and tissue microarrays. We found that the majority of precursor B/T lymphoblastic, follicular and diffuse large B-cell lymphomas, nodular lymphocyte-predominant Hodgkin lymphomas and angioimmunoblastic T-cell lymphomas strongly expressed the TOX protein. Burkitt and mantle cell lymphomas showed TOX expression in a small percentage of cases. TOX was not found in the majority of chronic lymphocytic leukemia, myelomas, marginal zone lymphomas and classical Hodgkin lymphomas. In conclusion, we describe for the first time the expression of TOX in normal and neoplastic lymphoid tissues. The co-expression of TOX and PD-1 identified in normal and neoplastic T cells is consistent with recent studies identifying TOX as a critical regulator of T-cell exhaustion and a potential immunotherapy target. Its differential expression may be of diagnostic relevance in the differential diagnosis of follicular lymphoma, the identification of the phenotype of diffuse large B-cell lymphoma and the recognition of peripheral T-cell lymphoma with a follicular helper T phenotype.

**Funding:** This work was supported by grants from the Plan Nacional de I+D+I, co-financed by the ISCIII-Subdirección General de Evaluación and the Fondo Europeo de Desarrollo Regional (FEDER), CIBERONC - CB16/12/00291 (MAP), and Programs of R&D activities among research groups of the Community of Madrid in Biomedicine (B2017/BMD-3778) (MAP, GR, JFGG). All funders had no role in study design, data collection and analysis, decision to publish, or preparation of the manuscript.

**Competing interests:** The authors have declared that no competing interests exist.

# Introduction

Thymocyte selection-associated high-mobility group box (TOX) is a member of a small subfamily of proteins (TOX2, TOX3 and TOX4) that share almost identical High Mobility Group (HMG)-box sequences and are highly conserved between mice and humans [1]. HMG proteins contain DNA-binding domains that allow them to produce specific changes in target DNA structure and modulate the formation of multi-protein complexes [2]. Using gene microarray technology, Wilkinson *et al.* first described *TOX* as a thymic transcript that was highly upregulated in CD4/CD8 double positive thymocytes and downregulated in mature CD4$^+$ cells [3].

Forced expression of Tox in the thymus of transgenic mice changed the differentiation program of developing T cells, suggesting its involvement as a key player in differentiation during lymphocyte development [4]. Tox-deficient mice have a severe block at a transitional stage of positive selection in the thymus, leading to loss of the CD4$^+$ T lymphocyte lineage [4]. Also Tox$^{-/-}$ mice display a failure of lymph node organogenesis and a drastic reduction in the frequency and size of Peyer's patches, suggesting it might also be involved in germinal center (GC) [5] B lymphocyte development and function [4]. In addition, TOX is highly expressed during *in vitro* natural killer (NK) differentiation and down-regulation of TOX decreased the population of NK cells [6, 7]. Recently, six studies have identified TOX as critical transcriptional and epigenetic coordinator of CD8$^+$ T-cell exhaustion in response to T-cell receptor stimulation and NFAT activation in infection and cancer [8–13]. These studies identify TOX as a central player in the regulation of T-cell responses and a future immunotherapeutic target.

In a previous gene expression profiling study, we identified a specific gene signature upregulated in follicular lymphomas (FL) and downregulated in a large proportion of nodal marginal zone lymphomas (NMZL) [14]. In a following analysis of the data obtained from this study, we found that TOX was expressed in multiple B-cell lymphoma types including a high proportion of large B-cell lymphoma cases. Furthermore, Schrader *et al* reported that TOX was expressed in both reactive and neoplastic GC B cells such as those in primary cutaneous follicle center lymphoma (PCFCL), secondary cutaneous FL and in a proportion of BCL6$^+$ primary cutaneous large B-cell lymphomas [15].

Further support for the possible diagnostic relevance of TOX expression is provided by recent reports showing that the *TOX* gene family is aberrantly expressed or mutated in diverse types of lymphoma and other cancer types [13, 16–19]. Notably, overexpression of TOX was found to have adverse prognostic implications in cutaneous T-cell lymphomas (CTCL), where it correlated with disease progression and mortality [16].

To further investigate the role of TOX and its potential diagnostic value in lymphomas here we have evaluated the labeling of a new anti-TOX monoclonal antibody (mAb) that works on paraffin-embedded tissues from a large series of normal tissues and B- and T-cell neoplasms.

# Materials and methods

## mRNA expression of *TOX* across low-grade B-cell lymphomas

Gene expression data from lymph nodes with FL, lymph nodes with NMZL, spleens infiltrated by chronic lymphocytic leukemia (CLL), lymph nodes infiltrated by extranodal marginal zone lymphoma of mucosa-associated lymphoid tissue (MALT), lymph nodes infiltrated by splenic marginal zone lymphoma (SMZL), spleen infiltrated by mantle cell lymphoma (MCL) and reactive lymphoid tissue (8 lymph nodes and 7 spleens) were available from previous studies [14, 20]. A moderated Student t-test was used to determine that the *TOX* gene was differentially expressed in FL (q-value <0.05; absolute fold-change >2.0) [21]. Pearson's correlation

was performed to identify the top-200 genes that positively and negatively correlated with *TOX* expression. Gene-set enrichment analysis (GSEA [22]) comparing NMZL versus FL and enrichment map have been additionally performed as described [23]. Further details are provided in Supporting Information (SI) (S1 Text, S1, S2 and S3 Figs and S1 Table in S1 Text).

## Production of an anti-TOX monoclonal antibody

A new anti-TOX mAb (clone NAN448B) was produced by immunizing Wistar rats with the amino terminal 250 residues of TOX fused to a HIS-tag that was produced in the BL21 strain of *Escherichia coli*. Details of protein production, purification, rat immunization, hybridoma production and clone selection are available as S1 Text. The plasmid vectors used for antibody validation are described in S2 Table in S1 Text.

## TOX gene inactivation using CRISPR–Cas9 technology

TOX expression was silenced in the human MOLT4 cell line by CRISPR-Cas9 technology. The third exon of the human *TOX* gene (ENSG00000198846) was analyzed looking for PAM sequences (NGG). The online MIT webtool (http://crispr.mit.edu/) was used to filter the best candidates, avoiding those with high numbers of off-target sequences and those with repetitive nucleotides. Two different guide RNAs (gRNAs) were designed within the aforementioned exon (sgTOX1_1: GGTGCACCAGCGAGTGGTCT, sgTOX1_2: AGCAGGCCATTATGGTTCAT). The gRNAs were cloned into a previously used lentiviral backbone (pLV-CRISPR) [24]. Details for virus production and MOLT4 transduction are provided in the S1 Text.

## Western blot

Western blot (WB) analyses of TOX protein were performed using total protein extracted from 19 cell lines lysed in a RIPA lysis buffer (Millipore, USA) with protease inhibitors (Roche, Germany). All detailed methodology is described in the S1 Text. Blotting membranes were incubated overnight with blocking solution (5% milk in PBS) and immunoblotted for 1 h at room temperature with anti-TOX NAN448B mAb (diluted 1:200), and anti-vinculin monoclonal antibody (diluted 1:100000), followed by incubation with HRP-conjugated secondary antibody (DAKO, Glostrup, Denmark). Information about commercial antibodies used is showed in S3 Table in S1 Text.

**Table 1. Expression of TOX in normal human tissues.**

| Hematopoietic | |
|---|---|
| Tonsil | T-cells in the interfollicular areas and B and T cells within the GC |
| Thymus | Lymphoblast in the thymic cortex; GC B-cells and T cells in the medulla |
| Spleen | White pulp GC B and T cells and scattered cells in the red pulp |
| Bone marrow | Isolated stromal cells and small lymphocytes |
| **Non-Hematopoietic** | |
| Intestine | Undifferentiated crypt cells, also myenteric plexus |
| Lung | Isolated pneumocytes |
| Kidney | Distal convoluted tubule |
| Ovary | Functional stroma |
| Pancreas | Endocrine cells |
| Testis | Sertoli cells |

## Human and mouse samples and cell lines

Labeling with the TOX mAb was performed across five types of reactive lymphoid tissues: lymph node, tonsil, bone marrow, thymus and spleen and 227 different lymphomas corresponding to 20 different subtypes (Tables 1 and 2). All the normal and tumor samples were retrospectively collected from the Biobank HUMV-IDIVAL and Biobank IIS-FJD, in accordance with the technical and ethical procedures of the Spanish National Tumor Bank Network, including patients informed written consent and double coded anonymization process. This project was approved by CEIC—FJD (Jiménez Díaz Foundation) on 06/26/2018 (minutes n° 10/18 with code PIC075-18_FJD).

Tissue samples for the validation of Tox mAb in mouse tissue were collected from male C57BL/6 mice. Animal experiments were performed under the experimental protocol approved by the Institutional Committee for Care and Use of Animals from Consejería de Medio Ambiente y Ordenación del Territorio of the Comunidad de Madrid (Madrid, Spain) with reference numbers 10/048238.9/15 and 10/251450.9/14 and reference projects PROEX038/15 and PROEX232/14. All efforts were made to minimize animal suffering.

DB, SUDHL4, SUDHL10, DOHH2, RAMOS, DAUDI, RAJI, OPM2, U266, YT, SUPT1, MOLT4, HH and WSU-NHL cell lines used in the present study were obtained from the German Collection of Microorganisms and Cell Cultures (DSMZ, Braunschweigh, Germany).

**Table 2. TOX expression in lymphomas.**

|  | No. cases | Positive cases | Negative cases | % Positive cases |
|---|---|---|---|---|
| Precursor Neoplasms |  |  |  |  |
| Precursor B lymphoblastic lymphoma | 3 | 2 | 1 | 66% |
| Precursor T lymphoblastic lymphoma | 7 | 5 | 2 | 71% |
| Mature B-cell neoplasms |  |  |  |  |
| Chronic lymphocytic leukemia | 16 | 1 | 15 | 6% |
| Follicular lymphoma | 52 | 34 | 18 | 65% |
| Mantle cell lymphoma | 14 | 2 | 12 | 14% |
| Diffuse large B-cell lymphoma |  |  |  |  |
| GC Type | 18 | 16 | 2 | 89% |
| Non-GC type | 14 | 7 | 7 | 50% |
| Burkitt lymphoma | 15 | 5 | 10 | 33% |
| Marginal Zone lymphomas |  |  |  |  |
| Nodal marginal zone lymphoma | 8 | 1 | 7 | 12% |
| MALT lymphoma | 4 | 0 | 4 | 0% |
| Splenic marginal zone lymphoma | 9 | 1 | 8 | 11% |
| Myeloma | 5 | 0 | 5 | 0% |
| Mature T-cell neoplasms |  |  |  |  |
| Peripheral T-cell lymphoma | 9 | 2 | 7 | 22% |
| Angioimmunoblastic T-cell lymphoma | 12 | 10 | 2 | 83% |
| Anaplastic large cell lymphoma ALK+ | 4 | 1 | 3 | 25% |
| Mycosis fungoides | 5 | 5 | 0 | 100% |
| Hodgkin lymphomas |  |  |  |  |
| Nodular lymphocyte-predominant HL | 10 | 9 | 1 | 90% |
| Lymphocyte-rich cHL | 3 | 0 | 3 | 0% |
| Nodular sclerosis cHL | 11 | 0 | 11 | 0% |
| Mixed cellularity cHL | 8 | 2 | 6 | 25% |

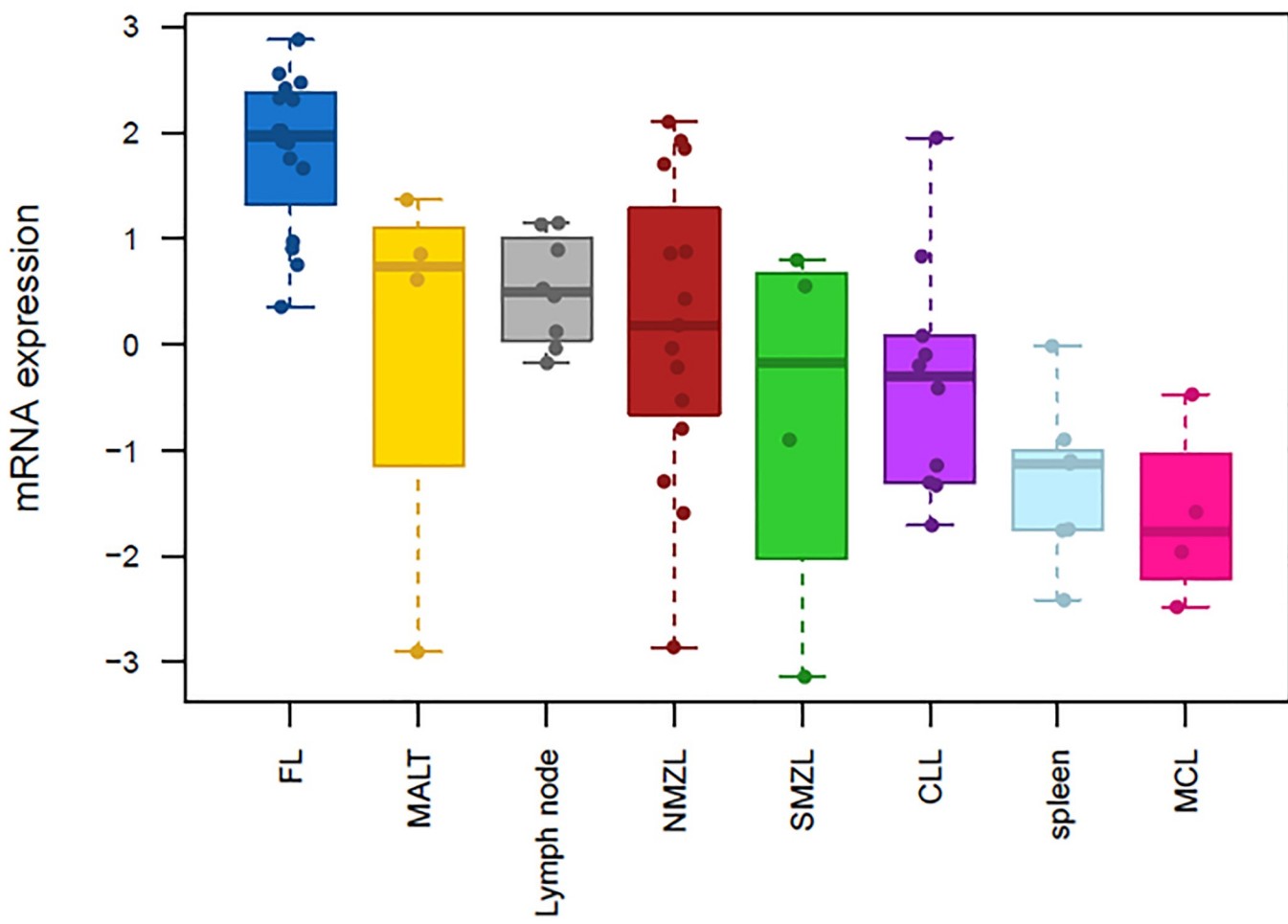

**Fig 1. Boxplot on mRNA expression of TOX across low-grade B-cell lymphomas and reactive lymphoid tissues.** Relative gene expression of *TOX* mRNA expression across B-cell lymphomas and lymphoid reactive tissues (n = 72): 16 lymph nodes with FL (blue), 4 lymph nodes infiltrated by MALT lymphoma (yellow), 8 reactive lymph nodes (grey), 15 lymph nodes with NMZL (red), 4 lymph nodes infiltrated by SMZL (green), 14 spleens infiltrated by CLL (light blue), 7 reactive spleens and 4 spleens infiltrated by MCL (magenta). Statistical significance was obtained when TOX expression in FL was separately compared to each of the other histotypes (FDR <0.05, fold-change >2.0. S1 Table in S1 Text). Expression data from each individual case is illustrated by single dots.

TOLEDO, AKATA, MYLA, KARPAS299 and HEK293T cell lines were kindly provided by Miguel Angel Piris from Department of Pathology, Fundación Jiménez Díaz, Spain. All the cell lines were authenticated by Short tandem repeat (STR) profiling. Cell line authentication reports are available in the S7 Fig in S1 Text.

### Immunohistochemistry, double immunoenzymatic and double immunofluorescence

Formalin-fixed, paraffin-embedded (FFPE) tissues from many of the normal and tumor samples were included in several tissue microarray (TMA) blocks using a Tissue Arrayer Device (Beecher Instrument, Silver Spring, MD, USA). IHC analyses were performed on TMAs or full tissue sections. Antibody sources are described in S3 Table.

The Bond Polymer Refine detection system (Leica Biosystems, Germany) was used for single and double immunoenzymatic labeling of FFPE tissues. Double labeling and double

immunofluorescence techniques were performed to assess the relationship between TOX and CD20, CD10, BCL6, CD3, CD4, CD8, Ki67, PD1, CD30, CD68, IgD and CD138. Protocols and antibody details are available in the S1 Text.

## Results

### Gene expression profiling identifies *TOX* overexpression in FL and low expression across low-grade B-cell malignancies and reactive lymphoid tissues

GEP analysis revealed significant overexpression in FL of *TOX* mRNA across low-grade B-cell lymphoma samples and reactive lymphoid tissues. In particular, significant *TOX* overexpression was found in FL when compared to NMZL (3.14 fold, FDR <0.05) [14]. *TOX* gene expression in FL was then further analyzed in comparison to other B-cell lymphomas (MALT, SMZL, MCL and CLL) and reactive lymphoid tissue (lymph node and spleen), where we found significant *TOX* overexpression in FL for all tests (fold change >2, FDR <0.05. Fig 1, S1 and S2 Tables Fig in S1 Text).

Consistent with the *TOX* mRNA overexpression in FL, enrichment analyses showed that the top-200 genes positively correlating with *TOX* significantly overlapped (Mann-Whitney, p <0.001) with up-regulated signatures in FL (compared either to NMZL or reactive lymph node); GC B-cell signatures, either normal or tumoral (GCB-like DLBCL); and proliferation pathways in B-cell lymphoma. Conversely, the top-200 genes that negatively correlated with *TOX* exhibited significant overlap with post-GC B-cell signatures, including down-regulated signatures in FL (compared to reactive lymph node or NMZL either), IRF4 programme in myeloma and plasma cells and MYD88 up-regulated transcriptome (S3 Fig in S1 Text).

### TOX antibody validation

A rat mAb (clone NAN448B, isotype IgG1/k) was raised to the N-terminus of TOX. To confirm lack of cross-reactivity with the other TOX family members, immunohistochemistry (IHC) was performed on frozen cytospin preparations of MYC-tagged human TOX and GFP-tagged TOX2, TOX3 and TOX4 proteins expressed in HEK293T cells. Labeling with the anti-MYC and GFP mAbs confirmed the efficiency of transfection and NAN448B specifically detected only TOX (Fig 2A). The same results were confirmed by WB (Fig 2A).

The specificity of NAN448B mAb for the endogenous TOX protein was confirmed by IHC in cytospin preparations of the MOLT4 cell line before and after *TOX* gene inactivation using CRISPR-Cas9 technology (TOX KO MOLT4). Strong nuclear expression of TOX was observed in wild type (WT) MOLT4 cells, while no staining was observed in TOX KO MOLT4 (Fig 2B). The same results were confirmed by WB (Fig 2B).

In the (WT) MOLT4 cell line and in the HEK-TOX two bands were detected at ~63 kDa and ~57kDa (Fig 2A and 2B). The 63kDa form was described previously [3] and it is presumed to be a post-translationally modified form of TOX. Neither band was found in MOLT4 after TOX depletion, confirming antibody specificity.

### TOX expression in human lymphoid cell lines

As shown in Fig 3, the 63kDa and 57kDa TOX proteins were observed in multiple lymphoma cell lines including DLBCL (DB, SUDHL4 and SUDHL10 (weak positivity)), BL (RAMOS and RAJI), FL (WSU-NHL), CTCL (HH and MYLA), TCL (YT) and T-ALL (SUP-T1 and MOLT4) cell lines. TOX was not expressed in DLBCL (TOLEDO and DOHH2), in BL

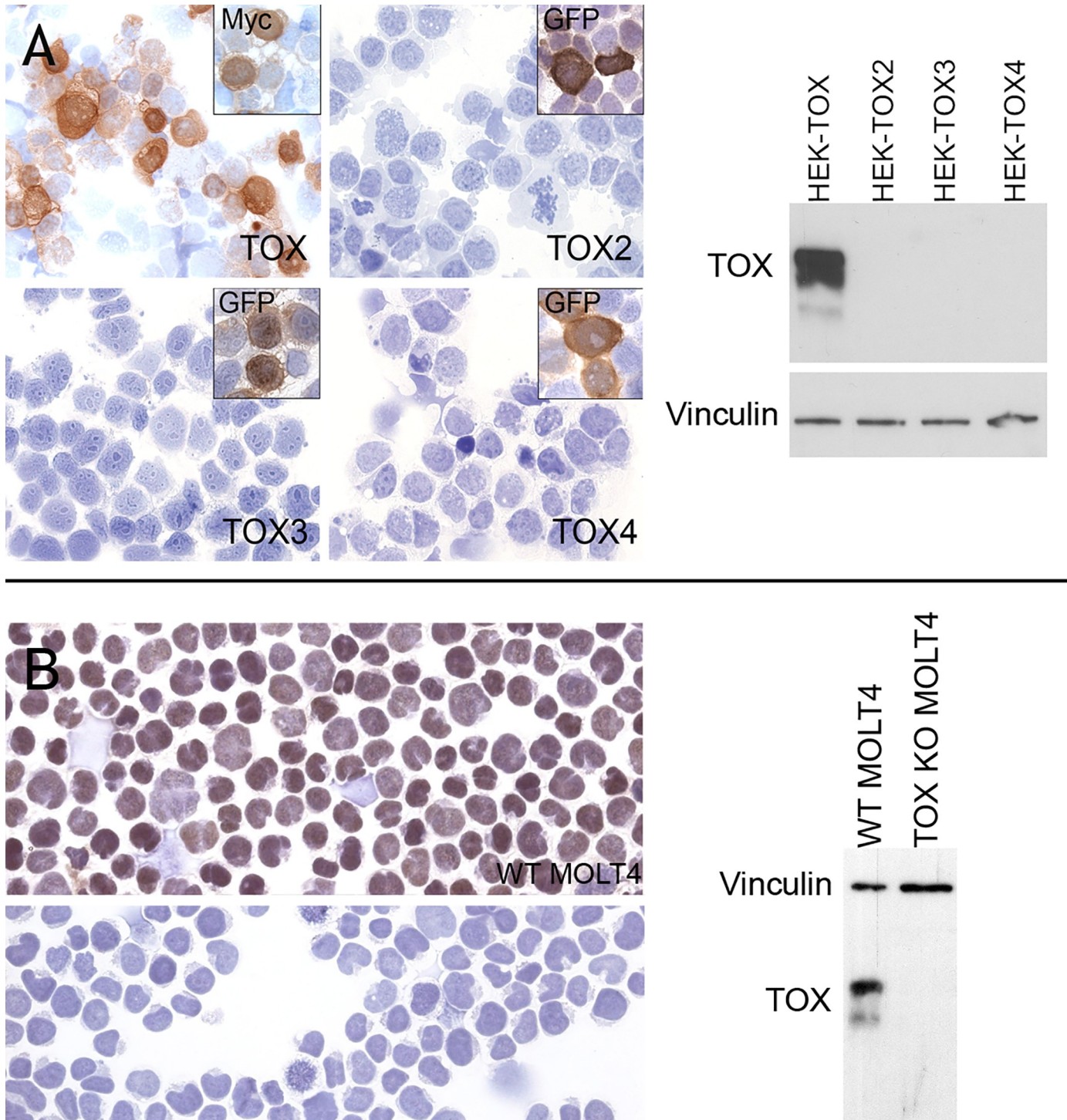

**Fig 2. Validation of the reactivity and specificity of the anti-TOX mAb NAN448B.** A) Immunocytochemical staining for TOX on cytocentrifuge preparations using NAN448B mAb. Staining of antibody NAN448B was observed in HEK-TOX-Myc transfectants but not in HEK-TOX2-GFP, HEK-TOX3-GFP and HEK-TOX4-GFP. The anti-Myc and anti-GFP mAbs were used to confirm transfection efficiency. Two bands of 63 kDa and 57kDa (weaker intensity) were detected by WB in HEK-TOX cell extracts while no expression was found in HEK-TOX2, HEK-TOX3 and HEK-TOX4. Anti-Vinculin antibody was used as loading control. B) Immunocytochemical staining for TOX on cytocentrifuge preparations of MOLT4 wild type (WT) and MOLT4 cell line after *TOX* gene inactivation using CRISPR-Cas9 technology (TOX KO MOLT4). Strong TOX staining was observed in the WT MOLT4 cytocentrifuge preparation while no staining was observed in the TOX MOLT4 KO cells. Two bands of

63 kDa and 57kDa (weaker intensity) were detected by WB in WT MOLT4 cell extracts while no expression was found in TOX KO MOLT4. Anti-Vinculin antibody was used as loading control.

(DAUDI and AKATA) cell lines. The myeloma (OPM2 and U266) and the anaplastic large cell lymphoma (KARPAS299) cell lines were also negative.

## TOX protein expression in normal human tissues

TOX protein expression in normal human tissues was analyzed using paraffin-embedded whole tissue sections. In reactive lymphoid tissues, TOX was detected in the nucleus of T cells in the interfollicular areas and in the nucleus and cytoplasm of GC B and T cells (Table 1 and Fig 4A). Strong TOX staining was present in CD4/CD8 double positive lymphocytes in the thymic cortex and in B and T cells in the medulla (Fig 4B). In spleen, TOX was present in the white pulp GC and in scattered cells in the red pulp. As with tonsil, the mantle zone lymphocytes in the spleen showed no staining (Fig 4C).

Using double immunoenzymatic and double immunofluorescence staining, TOX was highly expressed by CD20$^+$/CD10$^+$/BCL6$^+$ GC B lymphocytes (Fig 4D–4F) but was absent in IgD$^+$ B cells in the mantle zone (Fig 4N).

TOX was detected in a high number of CD3$^+$ T lymphocytes (Fig 4G) the majority of them being CD4$^+$ (Fig 4H) while only 5–10% were CD8$^+$ (Fig 4I).

A high number of Ki67 positive cells co-expressed TOX but a small population of Ki67$^+$TOX$^-$ cells was also observed. Interestingly all PD1$^+$ follicular helper T cells (TFH) strongly expressed TOX (Fig 4K). Double immunoenzymatic labeling highlighted TOX expression in CD30$^+$ activated B cells (Fig 4L) while CD68$^+$ macrophages, IgD$^+$ B cells and CD138$^+$ plasma cells located in the subepithelial areas lacked TOX expression (Fig 4M–4O).

TOX expression was also found in many cell types from other organs (Table 1 and S4 Fig in S1 Text) such as intestine (undifferentiated crypt cells and myenteric plexus), lung (Isolated pneumocytes), kidney (distal convoluted tubule) (S4 Fig in S1 Text), ovary (functional stroma), pancreas (endocrine cells) and testicle (Sertoli cells). Little to no staining was observed in breast, placenta, brain, liver, ovary and prostate.

Immunostaining using this antibody was also able to detect mouse Tox protein in murine paraffin-embedded tissue samples (S5 Fig in S1 Text). Tox expression in murine tissue was comparable to that in human tissues, with the exception of murine brain tissue where Tox was strongly expressed in many neurons of the cerebral cortex. Interestingly, we found high number of Tox$^+$ cells in mouse fetal liver suggesting that Tox could be expressed by hematopoietic stem cells (HSC) during murine development.

## TOX expression in B-cell lymphomas

The immunostaining results from paraffin sections of 227 primary lymphomas are summarized in Table 2. TOX was expressed in a large number of B-cell neoplasms, mostly those derived from GC B cells. In particular, TOX was expressed by 65% of FL (Fig 5A). Overall TOX expression was detected in 65% of FL and showed reduced frequency of expression in low grade 1 lymphomas (grade 1, 43%; grade 2, 73%; grade 3A, 73%; grade 3B, 67%). TOX was also expressed in a majority of the DLBCL cases analyzed (72%) (Fig 5B). When DLBCLs were classified as activated B-cell-like (ABC) or GC B-cell-like (GCB) accordingly to the Hans algorithm [26] the GC-DLBCL subtype showed increased TOX expression (GC: 89%; non-GC: 50%). More than half of the precursor B lymphoblastic lymphomas (B-LBL) expressed high levels of TOX protein (66%) (Fig 5C). Only 33% of BL cases expressed TOX.

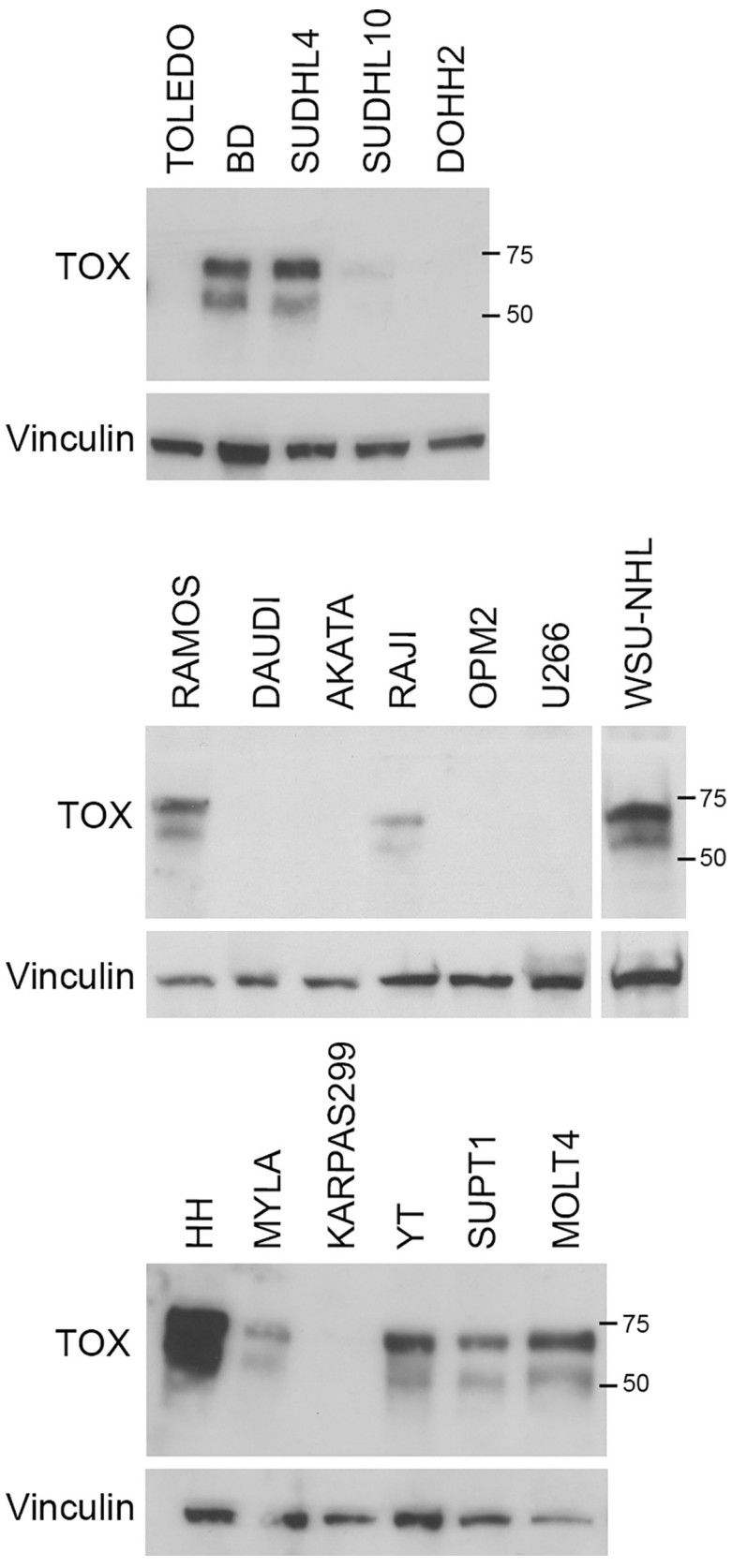

**Fig 3. TOX protein expression in lymphoid cell lines.** TOX expression in whole cell extracts of multiple tumor lymphoid cell lines were analyzed by WB using the NAN448B antibody. Anti-Vinculin antibody was used as loading control. The images represent three different gels.

In general, small B-cell lymphoma cases were rarely TOX-positive. Thus, TOX staining was seen only in 6% of CLL (Fig 5D), 14% of MCL, 12% of NMZL (Fig 5E) and 11% of SMZL. In these tissues, TOX staining was mostly negative with the exception of entrapped non-neoplastic GCs and T cells in the interfollicular area. None of the MALT lymphoma and myeloma cases expressed TOX. These data are in accordance with the absence of TOX protein in reactive plasma cells and in plasma cell derived cell lines tested by WB.

In HL, TOX recognized the neoplastic cells in almost all the nodular lymphocyte-predominant HL (NLPHL) cases (90%) (Fig 5F) while classical HL were mainly negative (Fig 5G), including nodular sclerosis (0%), lymphocyte rich (0%) and mixed cellularity (25%) subtypes. In NLPHL, the nuclear staining was moderate in the L&H (lymphocytic and histiocytic) cells while strong TOX positivity was also found in the PD1$^+$ rosette (red) surrounding the L&H cells (Fig 5F and 5G, inset image).

## TOX expression in T-cell lymphomas

In T-cell lymphomas, high TOX expression was found in all primary MF (100%) (Fig 5H) and in a high percentage of AITL (83%) (Fig 5I). To confirm the presence of TOX in AITL tumor cells, double immunoenzymatic staining was performed using an anti-PD1 mAb, confirming that all the neoplastic PD1$^+$ cells (red) also expressed TOX (Fig 5I, inset image). A small percentage of PTCL (22%) and ALCL (25%) expressed TOX.

## Discussion

TOX is a member of an evolutionarily conserved DNA-binding protein family required for development of T-cell subsets including CD4$^+$ effector T cells, regulatory T cells and NK T (NKT) cells [4]. While there are several reports describing the role played by TOX in development of the immune system, little is known about its expression and function in lymphomas.

Recently, gene expression profiling and additional immunohistochemical studies have suggested that TOX might represent a potential marker for the histological diagnosis of CTCL, being aberrantly expressed in CD4 neoplastic T cells in MF and Sezary syndrome [27]. Subsequent studies have described TOX expression in other CTCL types with a CD8$^+$ phenotype, thus limiting the diagnostic value of this marker in CTCL [17, 28]. Despite this, it seems that TOX could play an important pathogenic role in the context of CTCL tumor formation by enhancing the survival of malignant T cells [27].

In normal tissues high TOX expression was observed in CD4/CD8 double positive T lymphocytes in the cortex of the thymus, while in reactive lymphoid tissues it was strongly expressed by CD10+ and BCL6+ GC B cells. All PD1+ follicular helper T cells (TFH) outside and inside the GC were strongly TOX positive. Concerning the potential diagnostic clinical use of TOX immunostaining for cutaneous T-cell lymphomas, the presence of TOX-expression in reactive T cells present in inflammatory cutaneous disorders also prevents its diagnostic use for the recognition of MF. TOX protein expression was also found in non-lymphoid tissues such as intestine, lung, kidney, ovary, pancreas and testicle suggesting that TOX is involved in the differentiation of multiple cell lineages. Similar Tox expression patterns were found in murine tissues, with the exception of higher expression in brain and fetal liver.

In T-cell lymphomas high TOX expression was found in all the MF (100%), in precursor T lymphoblastic lymphoma (71%) and in a high percentage of AITL (83%) while PTCL and

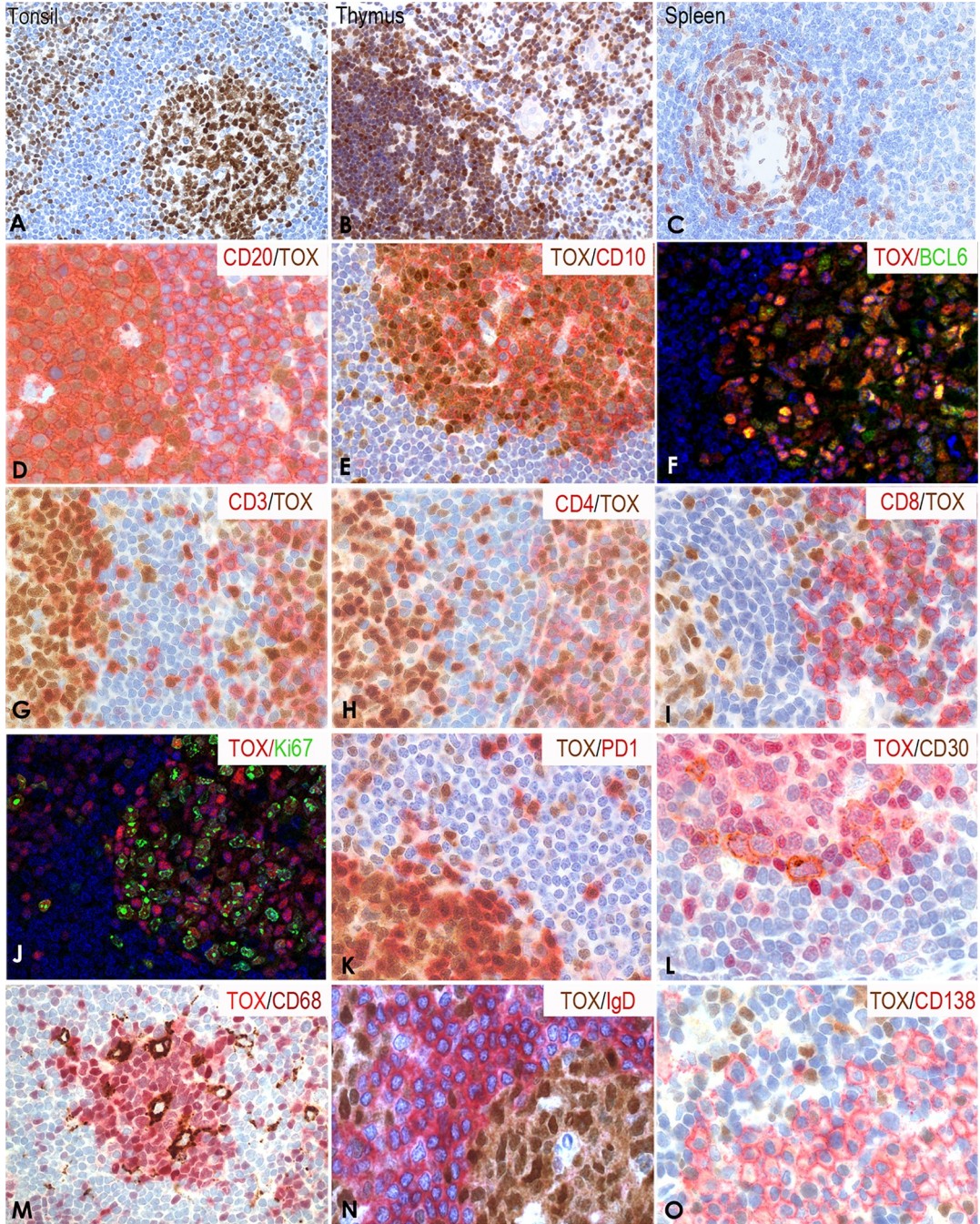

**Fig 4. Expression of TOX in normal lymphoid tissues.** (A) Single immunoperoxidase labeling in tonsil showed strong nuclear staining of TOX in GC B and T cells and in T cells in the interfollicular area. (B) TOX was also highly expressed by CD4/CD8 double positive T cells in thymus cortex and in proportion of mature B and T cells in the medulla. (C) In spleen, TOX was expressed by B and T cells in the white pulp GC and in scattered cells in the red pulp. Double immunenzymatic staining showed high expression of TOX [25] in CD20+/CD10+ cells (red) (D and E) in the GC and also in a proportion of B cells in the T-cell area. (F) Double immunofluorescence staining showed that the large majority of BCL6+ cells (green) also expressed TOX (red). High numbers of CD3+ T lymphocytes (red) (G) were TOX+ [25], the majority of them being CD4+ (red) (H) and only 5–10% were CD8+ (red) (I). Double immunofluorescence staining showed that a high number of Ki67 positive cells (green) co-expressed TOX (red) (J). All the PD1+ TFH (red) (K) and CD30+ activated B cells (L) [25] were TOX + (brown and red respectively). CD68 positive macrophages (M) [25], IgD + lymphocytes (red) (N) and CD138+ plasma cells (red) (O) lacked TOX staining.

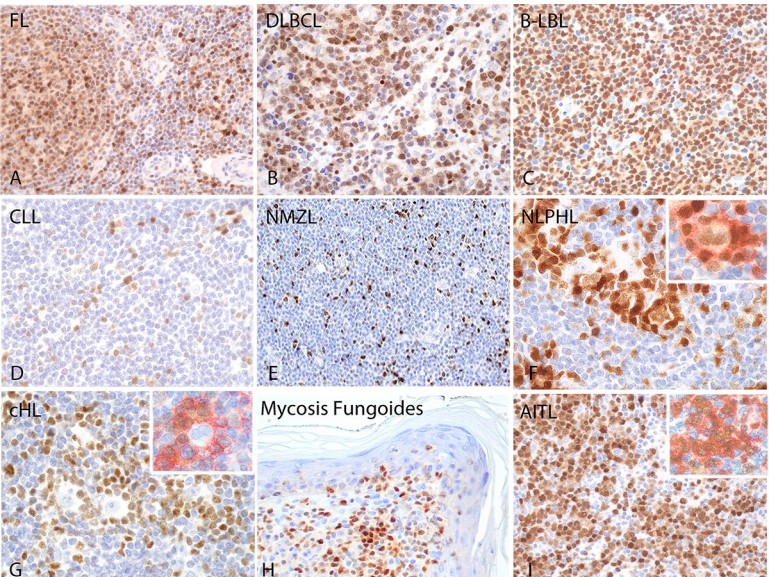

**Fig 5. TOX expression in B and T-cell lymphomas.** Strong TOX expression was found in the nuclei of tumor cells in FL (A), DLBCL (B) and precursor B-LBL (C). In CLL (D) and NMZL (E) TOX was found in the tumor microenvironment. In NLPHL (F) TOX was expressed by L&H cells and in the PD1+ rosette surrounding the tumor cells as shown by the double immunoenzymatic staining of TOX and PD1 (red) (upper image F and G). In cHL (G) TOX was found only in the tumor microenvironment. MF and AITL and MF also expressed high level of TOX (H and I). Double immunoenzymatic staining confirmed the expression of TOX [25] in PD1+ (red) in AITL tumor cells (inset image I).

ALCL were mainly negative. In AITL, TOX expression overlapped with PD-1 and other TFH markers demonstrating a strong relationship between TOX and PD-1 expression. Our data showing co-expression of TOX and PD-1 in both normal and neoplastic T cells and are consistent with a recent report showing that TOX reduces PD-1 degradation and promotes PD-1 translocation to the cell surface in CD8+ T cells, thus maintaining high PD-1 expression at the cell surface [13]. Alfei et al. also showed that Tox deficient CD8+ T cells showed a reduction in PD-1 expression and increased cytokine production [10]. Finally, high expression of TOX in peripheral CD8+ T cells correlated with poorer anti-PD-1 therapeutic responses and prognosis in hepatocellular carcinoma [13]. These studies highlight the importance of TOX in the detection of exhausted CD8+ T-cell populations in both cancer and infection, and in the response to anti-PD-1 targeted immunotherapy.

TOX was also expressed in reactive GC B cells and in primary cutaneous B-cell lymphomas (BCL6+ primary cutaneous large B-cell lymphoma leg-type and secondary cutaneous DLBCL), indicating that this gene may play a role in B-cell lymphoma pathogenesis [15]. Interestingly, genetic alterations (protein truncating mutation) of the *TOX* gene were recently found in DLBCL [29] and were predominantly in the high risk ABC subtype. A finding that could suggest a possible role of truncating mutations in the *TOX* gene as regulators of the plasma cell programme, as indicated by the absence of TOX expression in plasma cells and myelomas and the inactivation of this programme in ABC-DLBCL [30]. Another possibility is raised by the finding that double knockout of *Tox* and *Tox2* in chimeric antigen receptor (CAR) modified T cells infiltrating tumors affected the NF-κB pathway, including the increased accessibility of chromatin regions enriched for motifs that bind NF-κB [25].It is therefore also possible that suppressing TOX function may contribute to activation of NF-κB signaling, which is critical for the survival of the ABC-DLBCL subtype [25]. Further studies will be necessary to further

investigate these hypotheses. Interestingly *TOX* mutations were also found in primary central nervous system lymphoma (PCNSL), another large B-cell lymphoma type with an ABC phenotype [31, 32], where they have been found to be associated with shorter overall survival [33].

In B-cell lymphomas, high TOX protein expression was found in precursor B-LBL (66%), DLBCL (72%) and FL (65%) and in a low percentage of BL (33%). CLL, and myeloma did not (or only rarely) expressed TOX. The expression of TOX in FL correlates with grading, being more frequent in FL grades 2 and 3. We found that GCB-DLBCL cases (89%), showed preferential TOX expression compared with ABC-DLBCL (50%). In accordance with this observation and the *TOX* mutations in ABC-DLBCL, GEP analyses revealed high TOX expression associated with GCB-DLBCL signatures and proliferation cascades, while low expression correlated with post-GC B-cell signatures, and myeloma and plasma cell programs, also potentially suggesting a relevant role for TOX as a negative regulator of the late B-cell differentiation program. Further studies involving more extensive series of patients with available clinical data are warranted to establish whether TOX expression might improve clinical stratification of patients with mature B-cell malignancies. TOX expression was found in 33% of Burkitt lymphomas. This finding suggests that TOX expression could be restricted to a subset of GC-derived lymphomas. GC B cells are not a homogeneous population and this heterogeneity in the expression of GC related markers has previously been observed for other GC-related proteins such as GCET1/centerin and LMO2 [34, 35]. Additional mechanisms that could explain some variability in the expression of the TOX protein might be genetic alterations including copy number variation such as amplification/deletion or translocations involving TOX, as observed in DLBCL [25] and primary central nervous system lymphoma [31]. Further studies are required to elucidate the role played by TOX in Burkitt lymphomas.

Moreover, differences in TOX expression between FL and MZL were highlighted by both gene and protein expression, which might be relevant for differential diagnosis, especially in combination with other IHC markers. Our group has previously reported MNDA as a suitable marker for diagnosis of NMZL cases, with a very restricted expression in FL [36]. Thus, the combination of these two markers could add tools for the differential diagnosis of NMZL versus FL.

Finally, in HL, TOX recognized almost all the NLPHL (90%) with a staining that was specific for the L&H cells. TOX positivity was also found in the PD1$^+$ rosette surrounding the L&H cells. The combination of both characteristics might be useful for the differential diagnosis with classical HL.

In conclusion, we describe for the first time the expression of TOX in normal tissues and in a large series of B- and T-cell lymphomas. TOX represents a novel marker that may help in the identification of discrete B- and T-cell subpopulations, and in the differential diagnosis of GC versus non-GC DLBCL, FL vs MZL, NLPHL vs cHL and in the recognition of PTCL-TFH. Altered TOX function, either via physiological changes in protein expression or resulting from truncating somatic mutations, may also play a role in B-cell lymphoma pathogenesis. Importantly overcoming T-cell exhaustion in response to repeated antigenic stimulation is one of the most significant goals in immunotherapy, as it represents a key factor limiting the efficacy of T-cell responses to viral infections and tumor antigens. This TOX mAb will provide a valuable research tool to support further investigations into the role of TOX as a marker of exhausted CD8$^+$ T cells in both routine clinical samples and pre-clinical mouse models. Specific and reproducible detection of TOX$^+$PD1$^{high}$CD8$^+$ T cells should help to monitor the effectiveness of patients' T-cell immune responses to cancers and infectious agents, and the assessment of the benefits of novel immunotherapeutic approaches.

## Supporting information

**S1 Text. Supporting information.**
(DOC)

## Acknowledgments

The authors would like to thank all the members of the Tumor Bank Network and the Molecular Cytogenetics Unit of the CNIO for their technical contribution and assistance.

## Author Contributions

**Conceptualization:** Juan Fernando García-García, Santiago Montes-Moreno, Alberto J. Arribas, Alison H. Banham, Miguel Ángel Piris, Giovanna Roncador.

**Data curation:** Lorena Maestre, Juan Fernando García-García, Santiago Montes-Moreno, Alberto J. Arribas, Eduardo Caleiras, Alison H. Banham, Miguel Ángel Piris, Giovanna Roncador.

**Formal analysis:** Lorena Maestre, Alberto J. Arribas, Eduardo Caleiras, Miguel Ángel Piris, Giovanna Roncador.

**Investigation:** Patricia González-García.

**Methodology:** Lorena Maestre, Scherezade Jiménez, Ana Isabel Reyes-García, Álvaro García-González, Santiago Montes-Moreno.

**Software:** Alberto J. Arribas.

**Supervision:** Juan Fernando García-García, Santiago Montes-Moreno, Eduardo Caleiras, Miguel Ángel Piris, Giovanna Roncador.

**Validation:** Scherezade Jiménez, Ana Isabel Reyes-García, Álvaro García-González, Patricia González-García, Giovanna Roncador.

**Visualization:** Álvaro García-González.

**Writing – original draft:** Juan Fernando García-García, Alberto J. Arribas, Alison H. Banham, Miguel Ángel Piris, Giovanna Roncador.

**Writing – review & editing:** Juan Fernando García-García, Alison H. Banham, Miguel Ángel Piris, Giovanna Roncador.

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
