## [Decision Letter · Decision Letter 0]

11 Dec 2019

PONE-D-19-29134

High-mobility group box (TOX) antibody a useful tool for the identification of B and T cell subpopulation

PLOS ONE

Dear Dr Roncador,

Thank you for submitting your manuscript to PLOS ONE. After careful consideration, we feel that it has merit but does not fully meet PLOS ONE’s publication criteria as it currently stands. Therefore, we invite you to submit a revised version of the manuscript that addresses the points raised during the review process.

Please see reviewer comments below. Additionally, particular emphasis must be paid to the quality of the Western blot data and it's discussion within the text. For example, Fig.2A shows only a single band for TOX, why? Is it that HEK293 cells do not post-translationally alter the protein as suggested? I also note that reference 3 cited in the manuscript never ran TOX from their 293 cells on a Western, so this does not help clarify the observed anomaly. 

Please also note the journal's relatively new policy on cropped images, whereby the original, unmodified, images for all westerns and gels must now be submitted in the supplementary data.

Finally, the conclusions drawn from Fig.3 are invalid. You can not compare TOX expression between cell types whose loading controls are so vastly different. For example, to suggest RAJI express TOX while AKATA does not is simply unjustified. Indeed, loading controls are meant to be the same to show that direct comparisons can be made. I would ask the authors to repeat these Westerns to achieve the same loading levels before drawing any conclusions.

We would appreciate receiving your revised manuscript by Jan 25 2020 11:59PM. To enhance the reproducibility of your results, we recommend that if applicable you deposit your laboratory protocols in protocols.io, where a protocol can be assigned its own identifier (DOI) such that it can be cited independently in the future. For instructions see: http://journals.plos.org/plosone/s/submission-guidelines#loc-laboratory-protocols

We look forward to receiving your revised manuscript.

Kind regards,

Simon J Clark, D.Phil.

Academic Editor

PLOS ONE

Journal Requirements:

Please ensure that your manuscript meets PLOS ONE's style requirements, including those for file naming. The PLOS ONE style templates can be found at http://www.plosone.org/attachments/PLOSOne_formatting_sample_main_body.pdf and http://www.plosone.org/attachments/PLOSOne_formatting_sample_title_authors_affiliations.pdfThank you for including your ethics statement: "All the normal and tumor samples were retrospectively collected from the participant institutions, in accordance with the technical and ethical procedures of the Spanish National Tumor Bank Network, including informed consent and anonymization processes.Animal experiments were performed under the experimental protocol approved by the Institutional Committee for Care and Use of Animals. "a. Please amend your current ethics statement to include the full name of the animal ethics committee that approved your specific study.b. Once you have amended this/these statement(s) in the Methods section of the manuscript, please add the same text to the “Ethics Statement” field of the submission form (via “Edit Submission”).For additional information about PLOS ONE submissions requirements for ethics oversight of animal work, please refer to http://journals.plos.org/plosone/s/submission-guidelines#loc-animal-researchIn ethics statement in the manuscript and in the online submission form, please provide additional information about the patient records/samples used in your retrospective study. Specifically, please ensure that you have discussed whether all data/samples were fully anonymized before you accessed them and/or whether the IRB or ethics committee waived the requirement for informed consent. If patients provided informed written consent to have data/samples from their medical records used in research, please include this information.To comply with PLOS ONE submission guidelines please deposit the microarray data in a publicly available repository (you can find a list of repositories in the link here: https://journals.plos.org/plosone/s/data-availability#loc-recommended-repositories).As part of your revision, please complete and submit a copy of the ARRIVE Guidelines checklist for 1) rats used for immunisation and 2) C57BL/6 mice. This document aims to improve experimental reporting and reproducibility of animal studies for purposes of post-publication data analysis and reproducibility: https://www.nc3rs.org.uk/arrive-guidelines. Please include your completed checklist as a Supporting Information file. Please add to the methods section the ethical approval number(s) obtained for your study and the source of animals. Note that if your paper is accepted for publication, this checklist will be published as part of your article. PLOS ONE now requires that authors provide the original uncropped and unadjusted images underlying all blot or gel results reported in a submission’s figures or Supporting Information files. This policy and the journal’s other requirements for blot/gel reporting and figure preparation are described in detail at https://journals.plos.org/plosone/s/figures#loc-blot-and-gel-reporting-requirements and https://journals.plos.org/plosone/s/figures#loc-preparing-figures-from-image-files. When you submit your revised manuscript, please ensure that your figures adhere fully to these guidelines and provide the original underlying images for all blot or gel data reported in your submission. See the following link for instructions on providing the original image data: https://journals.plos.org/plosone/s/figures#loc-original-images-for-blots-and-gels.In your cover letter, please note whether your blot/gel image data are in Supporting Information or posted at a public data repository, provide the repository URL if relevant, and provide specific details as to which raw blot/gel images, if any, are not available. Email us at plosone@plos.org if you have any questions.Thank you for stating the following financial disclosure:  "The funders had no role in study design, data collection and analysis, decision to publish, or preparation of the manuscript.”   

" 

Please provide an amended Funding Statement that declares *all* the funding or sources of support received during this specific study (whether external or internal to your organization) as detailed online in our guide for authors at http://journals.plos.org/plosone/s/submit-now.  

Please state what role the funders took in the study.  If any authors received a salary from any of your funders, please state which authors and which funder. If the funders had no role, please state: "The funders had no role in study design, data collection and analysis, decision to publish, or preparation of the manuscript."

Additional Editor Comments (if provided):

Reviewers' comments:

Reviewer's Responses to Questions

**Comments to the Author**

1. Is the manuscript technically sound, and do the data support the conclusions?

Reviewer #1: Yes

2. Has the statistical analysis been performed appropriately and rigorously? 

Reviewer #1: Yes

3. Have the authors made all data underlying the findings in their manuscript fully available?

Reviewer #1: Yes

4. Is the manuscript presented in an intelligible fashion and written in standard English?

Reviewer #1: Yes

5. Review Comments to the Author

Reviewer #1: In the present study, the authors investigated TOX expression in normal and neoplastic

lymphoid tissues using a novel rat monoclonal antibody that recognizes its target

molecule in paraffin-embedded tissue sections. They found that the majority of precursor B/T lymphoblastic, follicular and diffuse large B-cell lymphomas, nodular lymphocyte-predominant Hodgkin lymphomas and angioimmunoblastic T-cell lymphomas strongly expressed the TOX protein. Burkitt and mantle cell lymphomas showed TOX expression in a small percentage of cases. TOX was not found in the majority of chronic lymphocytic leukemia, myelomas, marginal zone lymphomas and classical Hodgkin lymphomas. A co-expression of TOX in PD-1+ CD8 T cells suggested a possible role of TOX in T-cell exhaustion. The authors concluded that TOX differential expression may be of diagnostic relevance in the differential diagnosis of follicular lymphoma, the identification of the phenotype of diffuse large B-cell lymphoma and the recognition of peripheral T-cell lymphoma with a follicular helper T phenotype.

Please note the following:

- The authors reported expression of TOX in BL cell line RAMOS and RAJI, but this latter data was not clearly represented in figure 3

- The study demonstrated expression pf TOX in germinal center B cells, however Burkitt lymphoma resulted TOX negative in most of the cases, although it’ s a GC derived tumor. This data should be highlighted in the discussion section.

- 65% of FL turned out to be TOX+, the authors should report the percentage of positive cases in each grade (1/2, 3A, 3B)

6. PLOS authors have the option to publish the peer review history of their article (what does this mean?). If published, this will include your full peer review and any attached files.

Reviewer #1: No

---

## [Author Response · Author response to Decision Letter 0]

29 Jan 2020

Reply to the comment from academic editor 

Please find below all the answers to the academic editor and reviewer

Additionally, particular emphasis must be paid to the quality of the Western blot data and it's discussion within the text. For example, Fig.2A shows only a single band for TOX, why? Is it that HEK293 cells do not post-translationally alter the protein as suggested? I also note that reference 3 cited in the manuscript never ran TOX from their 293 cells on a Western, so this does not help clarify the observed anomaly. 

To address this point we have repeated the Western blot with 10µg of a cell extract of each TOX family protein (TOX, TOX2, TOX3, and TOX4) overexpressed in HEK293T cells. Using increasing concentrations of the lysate from HEK293T cells expressing recombinant TOX 

(2.5µg, 5µg and 10µg) we demonstrate the presence of the second lower molecular weight form of TOX (weaker intensity) when the protein concentration is increased (5µg and 10µg). 

We have included a Western blot with 10µg of the TOX HEK293T lysate in the manuscript.

Please also note the journal's relatively new policy on cropped images, whereby the original, unmodified, images for all westerns and gels must now be submitted in the supplementary data.

As requested by the editor, we have added the original and unmodified blotting images into the supplementary data. We have also added cell line authentication reports in the supporting information.

Finally, the conclusions drawn from Fig.3 are invalid. You cannot compare TOX expression between cell types whose loading controls are so vastly different. For example, to suggest RAJI express TOX while AKATA does not is simply unjustified. Indeed, loading controls are meant to be the same to show that direct comparisons can be made. I would ask the authors to repeat these Westerns to achieve the same loading levels before drawing any conclusions

As requested by the reviewer, we have repeated the Western blotting with similar levels of sample loading across the cell line panel. We have changed the Western blotting image in Figure 3 and added the uncropped original images in supporting information. 

Reply to the comment from reviewer

1. The authors reported expression of TOX in BL cell line RAMOS and RAJI, but this latter data was not clearly represented in figure 3

The data in Figure 3 has been replaced with new Western blotting data to achieve more equal sample loading across the cell line panel and this now clearly illustrates the presence of TOX expression in the Ramos and Raji cell lines.

2. The study demonstrated expression pf TOX in germinal center B cells, however Burkitt lymphoma resulted TOX negative in most of the cases, although it`s a GC derived tumor. This data should be highlighted in the discussion section.

TOX expression was found in 33% of Burkitt lymphomas. This finding suggests that TOX expression could be restricted to a subset of GC-derived lymphomas. GC B cells are not a homogeneous population and this heterogeneity in the expression of GC related markers has previously been observed for other GC-related proteins such as GCET1/centerin and LMO2 (1, 2). Additional mechanisms that could explain some variability in the expression of the TOX protein might be genetic alterations including copy number variation such as amplification/deletion or translocations involving TOX, as observed in DLBCL (3) and primary central nervous system lymphoma (4). Further studies are required to elucidate the role played by TOX in Burkitt lymphomas. 

This paragraph has been added to the main text.

1.Montes-Moreno S1, Roncador G, Maestre L, Martínez N, Sanchez-Verde L, Camacho FI, Cannata J, Martinez-Torrecuadrada JL, Shen Y, Chan WC, Piris MA.

Gcet1 (centerin), a highly restricted marker for a subset of germinal center-derived lymphomas.

Blood. 2008 Jan 1;111(1):351-8. 

2.Natkunam Y, Zhao S, Mason DY, et al. 

(Natkunam, 2007 #146). 

Blood. 2007;109(4):1636–1642. 

3.Schmitz R, Wright GW, Huang DW, Johnson CA, Phelan JD, Wang JQ, et al. 

Genetics and Pathogenesis of Diffuse Large B-Cell Lymphoma. 

N Engl J Med. 2018;378:1396-407. 

4.Cheng J, Tu P, Shi QL, Zhou HB, Zhou ZY, Zhao YC, et al. [Primary diffuse large B-cell lymphoma of central nervous system belongs to activated B-cell-like subgroup: a study of 47 cases]. Zhonghua Bing Li Xue Za Zhi. 2008;37:384-9. PMID: 19031717.

3. 65% of FL turned out to be TOX+, the authors should report the percentage of positive cases in each grade (1/2, 3A, 3B)

As suggested by the reviewer we have reported the percentage of positive cases in each grade of FL. Overall TOX expression was detected in 65% of FL and showed reduced frequency of expression in low grade 1 lymphomas (grade 1, 43%; grade 2, 73%; grade 3A, 73%; grade 3B, 67%).

These data have been added to the main text.

We also have amended our manuscript to meet all the Journal requirements

- We have included the full name of the animal ethics committee that approved our study in the supporting information and in the submission form.

- We have included additional information about the patient samples.

- Our GEP data was extracted from previous publications from our group (1-2)

The data from PMID 22110251 can be found in GEO GSE32233 while the data from PMID 23429603 are in process of being deposited in GEO repository. 

1.Arribas AJ, Gomez-Abad C, Sanchez-Beato M, Martinez N, Dilisio L, Casado F, et al. Splenic marginal zone lymphoma: comprehensive analysis of gene expression and miRNA profiling. Mod Pathol. 2013;26:889-901. PMID: 23429603.

2.Arribas AJ, Campos-Martin Y, Gomez-Abad C, Algara P, Sanchez-Beato M, Rodriguez-Pinilla MS, et al. Nodal marginal zone lymphoma: gene expression and miRNA profiling identify diagnostic markers and potential therapeutic targets. Blood. 2012;119:e9-e21. PMID: 22110251.

- All funders had no role in study design, data collection and analysis, decision to publish, or preparation of the manuscript.

---

## [Decision Letter · Decision Letter 1]

13 Feb 2020

High-mobility group box (TOX) antibody a useful tool for the identification of B and T cell subpopulation

PONE-D-19-29134R1

Dear Dr. Roncador,

We are pleased to inform you that your manuscript has been judged scientifically suitable for publication and will be formally accepted for publication once it complies with all outstanding technical requirements.

With kind regards,

Simon J Clark, D.Phil.

Academic Editor

PLOS ONE

Additional Editor Comments (optional):

Reviewers' comments:

Reviewer's Responses to Questions

**Comments to the Author**

1. If the authors have adequately addressed your comments raised in a previous round of review and you feel that this manuscript is now acceptable for publication, you may indicate that here to bypass the “Comments to the Author” section, enter your conflict of interest statement in the “Confidential to Editor” section, and submit your "Accept" recommendation.

Reviewer #1: All comments have been addressed

2. Is the manuscript technically sound, and do the data support the conclusions?

Reviewer #1: Yes

3. Has the statistical analysis been performed appropriately and rigorously? 

Reviewer #1: N/A

4. Have the authors made all data underlying the findings in their manuscript fully available?

Reviewer #1: Yes

5. Is the manuscript presented in an intelligible fashion and written in standard English?

Reviewer #1: Yes

6. Review Comments to the Author

Reviewer #1: the authors responded adequately to the reviewer's suggestions, modifying figure 3 and expanding the results and discussion section

7. PLOS authors have the option to publish the peer review history of their article (what does this mean?). If published, this will include your full peer review and any attached files.

Reviewer #1: No

---

## [Editor Report · Acceptance letter]

20 Feb 2020

PONE-D-19-29134R1 

High-mobility group box (TOX) antibody a useful tool for the identification of B and T cell subpopulation 

Dear Dr. Roncador:

I am pleased to inform you that your manuscript has been deemed suitable for publication in PLOS ONE. Congratulations! Your manuscript is now with our production department. 

With kind regards,

on behalf of

Prof. Simon J Clark 

Academic Editor

PLOS ONE